# Early Skin-to-Skin Contact Does Not Affect Cerebral Tissue Oxygenation in Preterm Infants <32 Weeks of Gestation

**DOI:** 10.3390/children9020211

**Published:** 2022-02-06

**Authors:** Kathrin Hanke, Tanja K. Rausch, Runa Sosnowski, Pia Paul, Juliane Spiegler, Mirja Müller, Inke R. König, Wolfgang Göpel, Egbert Herting, Christoph Härtel

**Affiliations:** 1Department of Pediatrics, University of Lübeck, D-23538 Lüebeck, Germany; t.rausch@uni-luebeck.de (T.K.R.); Runa.Sosnowski@uni-wh.de (R.S.); Wolfgang_Goepel@uksh.de (W.G.); Egbert.Herting@uksh.de (E.H.); 2Institut für Medizinische Biometrie und Statistik, University of Lübeck, D-23538 Lübeck, Germany; inke.koenig@uni-luebeck.de; 3Department of Pediatrics, University of Würzburg, D-97080 Würzburg, Germany; Paul_P@ukw.de (P.P.); Spiegler_J@ukw.de (J.S.); Mueller_M30@ukw.de (M.M.); haertel_c1@ukw.de (C.H.)

**Keywords:** regional cerebral oxygenation saturation, near infrared spectroscopy, skin-to-skin contact, preterm infants

## Abstract

Aim: It was the aim of our study to determine the regional cerebral tissue oxygenation saturation (rcSO_2_) as an additional monitoring parameter during early skin-to-skin contact (SSC) in preterm infants with a gestational age of <32 gestational weeks. Methods: We conducted two observational convenience sample studies using additional monitoring with near-infrared spectroscopy (NIRS) in the first 120 h of life: (a) NIRS 1 (gestational age of 26 0/7 to 31 6/7 weeks) and (b) NIRS 2 (gestational age of 24 0/7 to 28 6/7 weeks). The rcSO_2_ values were compared between resting time in the incubator (period I), SSC (period II) and handling nursing care (period III). For the comparison, we separated the sequential effects by including a “wash-out phase” of 1 h between each period. Results: During the first 120 h of life 38/53 infants in NIRS 1 and 15/23 infants in NIRS 2 received SSC, respectively. We found no remarkable differences for rcSO_2_ values of NIRS 1 patients between SSC time and period I (95% confidence interval (CI) for the difference in %: SSC vs. period I [1; 3]). In NIRS 2, rcSO_2_ values during SSC were only 2% lower compared with period I [median [1. quartile; 3. quartile] in %; 78 [73; 82] vs. 80 [74; 85]] but were similar to period III [78 [72; 83]]. In a combined analysis, a small difference in rcSO_2_ values between SSC and resting times was found using a generalized linear mixed model that included gender and gestational age (OR 95% CI; 1.178 [1.103; 1.253], *p* < 0.0001). Episodes below the cut-off for “hypoxia”; e.g., <55%, were comparable during SSC and periods I and III (0.3–2.1%). No FiO_2_ adjustment was required in the vast majority of SSC episodes. Conclusions: Our observational data indicate that rcSO_2_ values of infants during SSC were comparable to rcSO_2_ values during incubator care and resting time. This additional monitoring supports a safe implementation of early SSC in extremely preterm infants in NICUs.

## 1. Introduction

Skin-to-skin contact (SSC) is an important means to support the growth and development of preterm infants, to stabilize the parent-child relationship and to reduce the risk for parental physical and mental health problems [1,2,3]. This is in line with benefits of a reduced sepsis and mortality risk and an increased likelihood of exclusive breast feeding [4,5]. Immediate SSC in the first days of life has been demonstrated to be feasible procedure. The implementation of early SSC, however, largely depends on the context of the neonatal intensive care unit, individual risk patterns of the infant (mechanical ventilation, small size for their gestational age) and the attitude of medical professionals, particularly considering the high vulnerability of extremely preterm infants [6,7,8]. Safety concerns around early SSC have been raised for the potential risk of hypothermia [9] and the occurrence of events of cardiorespiratory instability, e.g., episodes of bradycardia and desaturations with a potential impact on adverse short- and long-term outcomes of brain development [10,11]. In order to improve safety, additional monitoring of the regional cerebral oxygenation saturation (rcSO_2_) with near-infrared spectroscopy (NIRS) could be helpful, as demonstrated by the Safe-BOOS C trial [12] and other studies that suggest that rcSO_2_ might be a useful biomarker of brain vulnerability [13,14]. A previous report noted that SSC does not affect rcSO_2_ values in preterm infants not needing respirator support at later postnatal age [15], while rcSO_2_ monitoring in relation to early SSC during the first days of life has not yet been explored. In observational studies with infants at a gestational age of 26 0/7–31 6/7 and 24 0/7 to 28 6/7 weeks, respectively, we tested our hypotheses that (i) rcSO_2_ values are not different during SSC as compared with resting periods and (ii) rcSO_2_ values are lower during nursing care as compared with resting periods of infants.

## 2. Materials and Methods

### 2.1. Study Cohort

We enrolled 55 preterm infants with a gestational age of between 26 0/7 to 31 6/7 weeks between 1 November 2014 and 1 April 2016, prospectively, in the University of Lübeck Children’s hospital (NIRS 1) (Figure 1). In a second study (NIRS 2) we recruited 31 patients with a gestational age of between 24 0/7 and 28 6/7 weeks between 6 September 2016 and 30 January 2018. The rationale behind the recruitment of two study cohorts was to determine the value of rcSO_2_ monitoring in two independent cohorts with different vulnerability based on gestational age. The study protocols were not different between NIRS 1 and NIRS 2, apart from an additional data monitoring on gastrointestinal circulation measures in NIRS 2. The observational studies enrolled convenience samples based on the availability of neonatal staff and a timely approach to obtain informed consent from parents. The inclusion criteria was birth within a gestational age (26 0/7–31 6/7 and 24 0/7–28 6/7 weeks). The exclusion criteria was the presence of life threatening and congenital malformations. All infants were stabilized after birth according to NICU guidelines. Delayed cord clamping for 30–45 sec was performed as a routine procedure in all infants. Parents or legal representatives gave written informed consent about the participation in the study. The studies were approved by the ethics committee of the University of Lübeck (vote numbers, NIRS 1: 14–272 and NIRS 2: 16–225).

### 2.2. Monitoring Cerebral Oxygenation

The regional cerebral oxygenation was measured by NIRS (INVOS 5100 near infrared spectrometer, Somanetics Corp, Medtronic, Meerbusch, Germany). Therefore, a neonatal NIRS sensor was placed on the front-parietal side of the head. A transducer (light emitting diode and two distant sensors) and differential signals from both sensors revealed the venous-weighted percentage of oxygenated hemoglobin, i.e., (oxygenated hemoglobin/total hemoglobin). Data recording started at minute 5 of life after primary stabilization. The values for rcSO_2_ were recorded every 5–10 s. FiO_2_ and vital parameters such as heart rate, SpO_2_ and respiratory rate were documented every 10 min in the first hour of life and every two hours thereafter for at least 120 h of life. Blood pressure measurements were performed non-invasively three to 12 times per day.

### 2.3. Definitions

Incubator (resting) period (period I): The resting time in the incubator was defined as a period with no skin-to-skin contact and no care by neonatal staff

Skin-to-skin contact (SSC, period II): Skin-to-skin contact was defined as prone positioning of the preterm infant skin-to-skin on the mother’s or father´s chest outside the incubator.

Nursing care period (period III): Period III was defined as handling care, which usually occurred once per 8h-shift including temperature control; blood gas analysis, nurse care methods and physicians’ assessments (duration 45 min).

Wash-out period: Before and after each period I-III a “wash-out time” of 60 min was implemented.

### 2.4. Period Analysis

For the main illustration of the different periods, all measured longitudinal values of all children during all episodes of the different periods (period I, period II, and period III, respectively) were considered together for NIRS 1 and NIRS 2, respectively. To plot and test the different periods, for each child the median of all measured values during all episodes of the corresponding period was determined.

### 2.5. Statistical Analysis

Descriptive statistical analyses were performed for clinical parameters. For testing and visualization of the infants receiving SSC, the two studies were merged due to the overlapping gestational ages between the studies and the effect the gestational age has on the rcSO_2_ values. To test for differences in infants who received SSC, we used a generalized linear mixed model (GLMM) for the beta distributed response, with the corresponding logit link and the infant connected with the time interval as a random effect. The gender, gestational age and study were used as adjustment. Due to the limited value range of rcSO_2_ between 15% and 95%, the values of rcSO_2_ were re-scaled to the uniform measure. The model cannot handle values of 1 and therefore another transformation ((x × (*n* – 1) + 0.5)/n) was carried out, where n is the sample size and x the value to transform. For the prediction, the transformations were reversed. The type I error level was set to 0.05. To test the differences between pairs of periods with SSC, two tests were performed, and the Bonferroni-adjusted type I error level was accordingly set to 0.025 for an overall significance level of 0.05.

### 2.6. Software

We used the R version 4.1.2 (The R Foundation, Vienna, Austria) together with the SPSS 26.0 data analysis package (IBM Copl, New York, NY, USA) for all computations and visualizations. Plots were generated using the R package ggplot2 (3.3.5) and R-function glmmTMB together with predict.glmmTMB from package glmmTMB (1.1.2.3) for the generalized linear mixed model and the corresponding prediction.

## 3. Results

### 3.1. Clinical Characteristics of the Study Group

In NIRS 1 (*n* = 55) and NIRS 2 (*n* = 31), a total of 86 preterm infants were enrolled. Of these, 10 infants were excluded due to an absence of available rcSO_2_ values and 76 infants (NIRS 1: 53, NIRS 2: 23) were analyzed. The clinical characteristics are shown in Table 1. The median gestational age was 28.8 weeks with a median birth weight of 1149 g (NIRS 1: 29.7 weeks, 1270 g; NIRS 2: 26.9 weeks, 920 g).

### 3.2. Implementation of Skin-to-Skin Contact

SSC during the first 120 h of life was realized in 53 preterm infants, i.e., NIRS 1 38/53 (77%, 117 episodes) and NIRS 2 15/23 (65%, 43 episodes). The median duration of SSC was 98 (1. quartile; 3. quartile: [75; 120]) minutes in NIRS 1 and 120 [105; 135] minutes in NIRS 2. Notably, 14/76 preterm infants were primarily intubated and received invasive ventilation. In the subgroup of ventilated babies, 8 preterm infants had SSC in first 120 h of life.

### 3.3. Regional Cerebral Oxygenation Saturation Values Are Not Different between Resting Times and Nursing Care Periods

In the whole group of infants (NIRS 1: *n* = 53, NIRS 2: *n* = 23) considering all available rcSO_2_ values, nursing care (period III) and resting time (period I) did not reveal remarkable differences, i.e., NIRS 1 in %: 81 [74; 87] vs. 82 [76; 88] and NIRS 2 in %: 76 [71; 82] vs. 78 [71; 83]; Table 2).

### 3.4. Regional Cerebral Oxygenation Saturation during Skin-to-Skin Contact

In the subgroup of infants receiving SSC with available rcSO_2_ values (NIRS 1: *n* = 38, NIRS 2: *n* = 15), the median values for rcSO_2_ of all available rcSO_2_ values were not different between SSC and resting time (period I) in NIRS 1 patients (in %: 83 [78; 87] vs. 83 [77; 88], Table 3a). In NIRS 2, rcSO_2_ values were lower during SSC as compared with resting time in NIRS 2 patients (78 [73; 82] vs. 80 [74; 85]). In a combined analysis of both studies, the median values for rcSO_2_ showed a significant difference between SSC and resting times (median and quartiles in %: 81 [76; 86] vs. 82 [77; 87]; OR with 95% confidence interval from GLMM: 1.178 [1.103; 1.253], *p* < 0.0001; Table 3b,c, Figure 2). We observed no difference between SSC and nursing care (81 [76; 86] vs. 81 [76; 86]; 1.011 [0.930; 1.092], *p* = 0.7869). The predictions based on the GLMM for different gestational ages, gender and time periods are shown in Figure 2. Figure 3 demonstrates the intraindividual differences for each infant according to median values during the periods I-III. In addition, the time spent with rcSO_2_ values <55% during SSC was comparably low compared with resting time values (NIRS 1: 1.2 vs. 2.1%; NIRS 2: 0.4 vs. 0.3%, Table 4). To achieve these rcSO_2_ values, no adjustments to the FiO_2_ requirement were needed for most SSC episodes (NIRS 1: 101/117; NIRS 2: 29/43). The FiO_2_ needed adjusting to higher levels in 2/117 SSC episodes (NIRS 1) and 7/43 episodes (NIRS 2), while FiO_2_ was reduced during SSC in 6/117 episodes (NIRS 1) and 2/43 episodes (NIRS 2), respectively.

## 4. Discussion

Our prospective studies, including cerebral oxygenation monitoring during skin-to-skin contact, underline the feasibility and safety of early SSC during the first 120 h of life [7]. It is a particular strength of our approach to investigate highly vulnerable infants at an early stage of their cardiorespiratory adaptation. These data also provide a benchmark for improvement, as most immature babies with primary intubation have less opportunity to receive SSC in our context. We found a median 1% difference of rcSO_2_ values during SSC episodes and resting time periods, while hypoxic episodes (<55%) as per the definition of the Safe-BOOS-C trial were rare events [12]. In an exploratory analysis of resting periods versus nursing care periods no differences in rcSO_2_ values were demonstrated.

Early SSC is recommended for mothers and their healthy newborn infants [3]. SSC is an important measure to improve growth and development of infants, the parent –infant relationship, the rate of human milk feeding and parental health [4,5]. Additional monitoring such as NIRS might help to assure safety during SSC and to reduce events of hypoxia [10,11]. In our study cohort, 74% of preterm infants received SSC during the first 120 h of life. A Swedish study showed that SSC was documented in 64% of 520 infants with a birth weight <1000 g [7]. Hence, it is a major task to advocate early SSC among parents and health professionals, to improve the infrastructure of NICUs to allow early SSC for extremely preterm infants (even in the delivery room) and to even guide parents in the context of maternal health problems after delivery with limited resources [16,17].

The regional cerebral oxygenation saturation during skin-to-skin contact has been examined in a few previous studies with later timepoints of NIRS monitoring (day 8), a smaller sample size of preterm infants or a gestational age >28 weeks as compared to our cohort [15,16,18]. We additionally evaluated handling care as an observational period, which revealed mildly lower rcSO_2_ values as compared to resting time. We found no remarkable differences between SSC and handling care. Notably, our study cohort achieved a high rate of previously published target levels 55–85% during SSC [19,20,21]. These results support and reveal that skin-to-skin contact does not increase cerebral hemodynamic instability in the first 120 h of life, which is in contrast to previous observations by Bohnhorst et al. [11]. The authors describe an increase of hypoxic events during SSC during three 2 h NIRS recordings at a median age of 25.5 days.

Limitations of the study include single center design, convenience sample, differences in parents’ availability and the accuracy of the technology during movement. We did not correlate the rcSO_2_ values with the level of required intensive care of the individual infant (e.g., inotrope need, mechanical ventilation). In conclusion, our observational data indicate that the rcSO_2_ values of infants during SSC were comparable to rcSO_2_ values during incubator care and resting time. This additional monitoring supports a safe implementation of early SSC in extremely preterm infants, which should be advocated as an important measure in neonatal care.

## Figures and Tables

**Figure 1 children-09-00211-f001:**
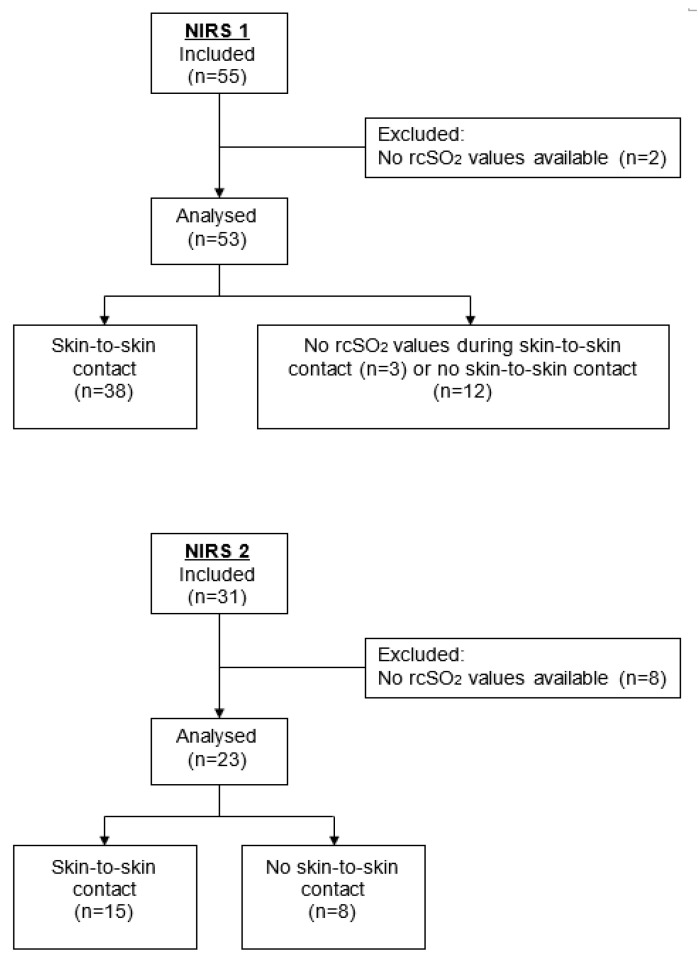
Flow diagram of the observational study cohorts.

**Figure 2 children-09-00211-f002:**
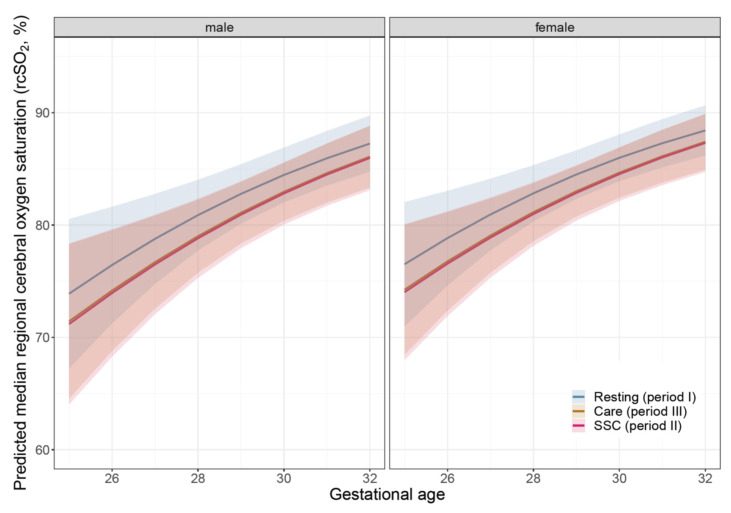
Prediction for median rcSO_2_ for gestational age based on gender and gestational age for resting (I), skin-to-skin contact (SSC, II) and nursing care (III) periods. Combined datasets of NIRS I and II are depicted. The areas around the lines correspond to the 95%-confidence interval for the prediction.

**Figure 3 children-09-00211-f003:**
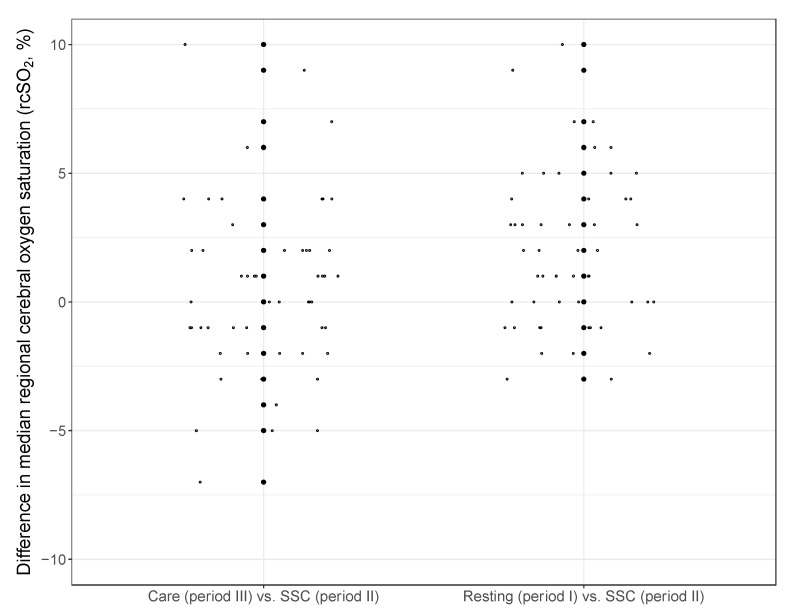
Intraindividual differences for all infants receiving SSC in the analysis of combined datasets (NIRS 1 and NIRS 2; *n* = 53). Values are based on the difference in the median of all measured values during the corresponding periods for each individual infant. Each small circle describes the difference of the infants between the period.

**Table 1 children-09-00211-t001:** Clinical characteristics of the observational study cohorts. Data are given as numbers (percentage) or median (quartile 1; quartile 3).

	NIRS 1	NIRS 2	Total
Number of infants	53	23	76
Gestational age (weeks)	29.7 [28.0; 31.2]	26.9 [25.3; 28.0]	28.8 [27; 30.7]
Birth weight (g)	1270 [1045; 1570]	920 [650; 980]	1149 [953; 1454]
Female gender	25 (47)	8 (35)	33 (43)
Multiple birth	17 (32)	2 (9)	19 (25)
Surfactant application	31 (58)	18 (78)	49 (65)
Oxygen application	31 (58)	18 (78)	49 (65)
Primary intubation	8 (15)	6 (26)	14 (18)
Duration of invasive ventilation (h)	25 [17.25; 58.75]	54.5 [10.25; 120]	25 [15.25; 77.5]
Pneumonia	2 (4)	2 (9)	4 (5)
Sepsis	15 (28)	3 (13)	18 (24)

**Table 2 children-09-00211-t002:** Regional cerebral oxygenation saturation (rcSO_2_) in % during period I (resting time) versus period III (nursing care). v is the number of available values in all infants, SD is standard deviation and Q is quartile.

	NIRS 1 (*n* = 53)		NIRS 2 (*n* = 23)	
Time Period	Period I Resting	Period III Care	Period I Resting	Period III Care
v	2,299,792	358,452	977,026	148,820
Mean ± SD	80.63 ± 9.79	79.45 ± 10.21	76.84 ± 10.06	75.45 ± 9.76
Median [Q_1_; Q_3_]	82 [76; 88]	81 [74; 87]	78 [71; 83]	76 [71; 82]
Minimum	15	15	15	15
Maximum	95	95	95	95

**Table 3 children-09-00211-t003:** a, b: Regional cerebral oxygenation (rcSO_2_) in % during period I (resting time) period II (skin-to-skin-contact; SSC) and period III (handling) for infants with SSC based on all available values (3a) or median values (3b) of infants during each period with an overview of the number of values per infant, which were used to evaluate the median value per infant. The results of the generalized linear model (GLMM) are shown in the last part (3c). Predictions using the estimate and standard error have to be rescaled for the 0–1-problem and the [15; 95] interval limitation. v is the number of available values in all infants, SD is standard deviation, Q is quartile, SE is standard error, OR is odds ratio and CI is confidence interval.

**a**	**NIRS 1 (*n* = 38)**			**NIRS 2 (*n* = 15)**		
**Time Period**	**Period I Resting**	**Period II SSC**	**Period III Care**	**Period I Resting**	**Period II SSC**	**Period III Care**
v	1,772,009	115,555	260,101	613,555	49,057	105,898
Mean ± SD	81.62 ± 9.56	81.62 ± 8.67	81.19 ± 8.85	79.48 ± 7.66	77.45 ± 7.45	77.44 ± 7.90
Median [Q_1_; Q_3_]	83 [77; 88]	83 [78; 87]	82 [76; 88]	80 [74; 85]	78 [73; 82]	78 [72; 83]
Minimum	15	25	19	15	15	15
Maximum	95	95	95	95	95	95
**b**	**Combined Data of NIRS 1 and NIRS 2**					
**Time Period**	**Period I Resting**	**Period II SSC**		**Period III Care**		
Median [Q_1_; Q_3_]	82 [77; 87]	81 [76; 86]		81 [76; 86]		
Minimum	68	62		67		
Maximum	95	92		95		
Number of values per infant in the periods						
Median [Q_1_; Q_3_]	41,910 [40,094; 45,362]	2812 [1260; 4506]		6770 [6511; 7386]		
Minimum	33,505	472		4698		
Maximum	60,814	9245		8447		
**c**	**Results of the GLMM**					
	**Estimate**		**SE**		**OR [95%-CI]**	***p*-Value**
Intercept	−3.0008		1.4369		0.050 [−2.767; 2.866]	
Female gender	0.1993		0.1626		1.220 [0.902; 1.539]	
Gestational age	0.1570		0.0485		1.770 [1.075; 1.265]	
NIRS 2	−0.1188		0.2051		0.888 [0.486; 1.290]	
SSC (period II)			Reference			
Resting (period I)	0.1639		0.0381		1.178 [1.103; 1.253]	<0.0001
Nursing care (period III)	0.0112		0.0413		1.011 [0.930; 1.092]	0.7869


**Table 4 children-09-00211-t004:** rcSO_2_ according to different thresholds including hypoxia and hyperoxia during period I (resting), period II (skin-to-skin-contact; SSC) and period III (handling) for infants with SSC.

	NIRS 1 (*n* = 38)			NIRS 2 (*n* = 15)		
rcSO_2_ (%)	Period I Resting	Period II SSC	Period III Handling	Period I Resting	Period II SSC	Period III Handling
v	1,772,009	115,555	260,101	613,555	49,057	105,898
15–< 55%	2.1	1.2	0.4	0.3	0.4	0.5
55–85%	60.3	63.4	65	78.8	87.5	84.7
15–< 65%	5.5	5.3	4.7	2.6	3.4	5.1
15–< 70%	9.7	9.5	10.7	9.6	13.7	14.7
15–< 80%	34.6	32.3	38.3	48.2	59.5	59.6
>85–95%	37.6	35.4	34.5	21	12.1	14.8

Data were presented as the percentage of rcSO_2_ values below, above, or within a threshold range. v Number of available values for all children in the period. rcSO_2_ below 15% or above 95% are not possible. >85–95% preterm infants with FiO_2_ >21% (NIRS 1: *n* = 31, NIRS 2: *n* = 18).

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
