# Peer review of "Early Skin-to-Skin Contact Does Not Affect Cerebral Tissue Oxygenation in Preterm Infants <32 Weeks of Gestation"

_children, 2022, doi:10.3390/children9020211_

Round 1

Reviewer 1 Report

Thank you for an important paper presenting results of NIRS of EPT and VPT infants during SSC. Some questions need to be answered or clarified in the manuscript to make it interpretable.

The paper publishes data from two studies(NIRS1 and NIRS2) with partly overlapping gestational ages. What was the rational behind the two recruitment periods and gestational ages? Was the study protocol the same otherwise?

Mean and median values for rcSO2 are presented. Why both? 

Were any adjustments done? If not, please state that data is unadjusted. Data is very hard to interpret without knowing the infants' cardiorespiratory state (intubated or non-invasive ventilation, inotropes, for example).

The flow diagrams in Fig 1 are different for NIRS1 and NIRS2; in NIRS2 it appears that infants with no SSC (n=8) were also analysed. How? If so, what were the characteristics of these infants in terms of gestational age etc?

Please add units in Table 1 (median gestational age?)

Please explain what "v" is, in Table 2.

In Table 3, please explain the difference between a and first section of b

Author Response

Dear Editor,

thank you very much for your decision letter regarding our manuscript „Early skin-to-skin contact does not affect cerebral tissue oxygenation in preterm infants < 32 weeks of gestation“ by Hanke et al. (Children 1524595). We specifically thank the reviewers for their helpful advices and did an earnest attempt to revise the manuscript accordingly.

Reviewer 1:

1)Thank you for an important paper presenting results of NIRS of EPT and VPT infants during SSC. Some questions need to be answered or clarified in the manuscript to make it interpretable.The paper publishes data from two studies (NIRS1 and NIRS2) with partly overlapping gestational ages. What was the rational behind the two recruitment periods and gestational ages? Was the study protocol the same otherwise?

Ad 1) The rationale behind the recruitment of two study cohorts was to determine the value of rcSO2 monitoring in two independent cohorts of different vulnerability based on gestational age. The study protocols were not different between NIRS 1 and NIRS 2 apart from an additional data monitoring on gastrointestinal circulation measures in NIRS 2.

2) Mean and median values for rcSO2 are presented. Why both? 

Ad2) We presented mean, median and quartiles in ordert o account for interindividual variability.

3) Were any adjustments done? If not, please state that data is unadjusted.

Ad 3) Gender, gestational age, and study were used as adjustment in mixed models as described under Methods.  

4) Data is very hard to interpret without knowing the infants' cardiorespiratory state (intubated or non-invasive ventilation, inotropes, for example).

Ad 4) We did not correlate the rcSO2 values with the level of required intensive care of the individual infant (e.g., inotrope need, mechanical ventilation). We stated this as limitation.

  1. The flow diagrams in Fig 1 are different for NIRS1 and NIRS2; in NIRS2 it appears that infants with no SSC (n=8) were also analysed. How? If so, what were the characteristics of these infants in terms of gestational age etc?

Ad 5) All infants including those without SSC were analysed for differences between resting period and nusring care period. The subgroups of infants with SSC during the first 120 hours of life were analysed for differences between all three periods.

  1. Please add units in Table 1 (median gestational age?). Please explain what "v" is, in Table 2. In Table 3, please explain the difference between a and first section of b.

Ad 6) Revisions were made accordingly.

Reviewer 2 Report

The study investigates cerebral tissue oxygenation in preterm infants as a biomarker for brain vulnerability. It is used to determine the safety of skin-to-skin contact during the first 120h of life in preterm infants. Therefore, cerebral tissue oxygenation is compared between periods of resting time, skin-to-skin contact, and nursing care. The study has the potential to alleviate concerns in the medical field towards early skin-to-skin contact, as the data suggests no substantial differences in cerebral tissue oxygenation. Given the multitude of beneficial effects of early skin-to-skin contact, the study provides a significant contribution to the field.

Yet, there are some aspects which need to be addressed, particularly concerning the statistical analysis:

  1. I was wondering why you did not choose to combine the data of both studies. Besides the gap in data collection, I could not identify any meaningful differences in the data collection for both studies. The gestational age in NIRS 2 was slightly lower, but as you do not investigate the effect of gestational age, I would not see this as a reason not to combine the data. Should I have missed the reason why the data cannot be combined, please explain it in more detail.
  2. You explore the difference in regional cerebral oxygenation saturation between resting time and handling care. While this is an interesting and important question in my opinion, it is not introduced in the introduction. The topic should at least be added as an exploratory hypothesis, stating that you will look at the comparison, even though you have no clear expectations concerning it.
  3. The statement in Heading 3.3 is different from your statement within the text. As you do not test the difference between both conditions, I do not see the statement in Heading 3.3 justified by the analysis. If you did test it, please report the results fully.
  4. What is given in parenthesis in Table 1? I’m assuming the numbers in brackets are confidence intervals, but neither is specified.
  5. In my opinion, the chosen statistical methods are not appropriate for the data structure or the research question. When you’re using all the available data, you do not currently account for dependencies in the data. When you only use the median, you lose the majority of your power. In addition, classical significance testing cannot be used to make statements such as “There is no difference in rcSO2 between SSC and resting”, you can only say “The hypothesis that no difference in rcSO2 between SSC and resting exists cannot be rejected”, which is not the same. I also don’t understand why you use non-parametric tests designed for non-normally distributed data. Theoretical considerations and your descriptive statistics suggest that your dependent variable is normally distributed. Given your hypothesis and your data structure, I would suggest Bayesian multilevel models (also called mixed models). The advantage here is that all your available data could be included, while also controlling for the dependencies in the data. Your data is hierarchical with two levels, assessments at Level 1 and participants at Level 2. The time period (incubator vs. SSC vs. care) is a Level 1 predictor as a characteristic of the assessment. And you could also control for aspects like gestational age, which is a Level 2 predictor as a characteristic of the participant (if you add this predictor to your analysis, z-transform it first as a version centering at the grand mean, see Enders & Tofighi, 2007). Both and more possible confounding variables (e.g., gender) could be included in the analysis at the same time.
    As you list SPSS as a software you are already using, I would suggest JASP to conduct Bayesian multilevel models. It is as easy to use and openly available without charge. It also offers reasonable priors which you can use, should you not want to specify your own. You can use “Bayesian Linear Mixed Models” if your data is normally distributed as I think and “Bayesian Generalized Linear Mixed Models” if they are not. JASP will automatically dummy code your time period variable for the analysis.
    The advantage with this method over yours is that you’ll receive credibility intervals, which can be interpreted to make statements as “There is no difference in rcSO2 between SSC and resting”. You also do not lose any power and you can address your hypothesis with one test instead of multiple. You also can use your data as it is, without adding an artificial error.
  6. Should you not be able to change the statistical methods, here are some more detailed comments concerning your current analysis:
    1. 3, lines 101-103: Please provide more information on the distribution of the added error. Was this additional error accounted for when interpreting the results?
    2. 3, lines 103-107: The Bonferroni correction is only necessary when you use multiple tests to test the same hypothesis. This only applies to four Wilcoxon-signed-rank tests here: rest vs. SSC in NIRS 1, care vs. SSC in NIRS 1, rest vs. SSC in NIRS 2, care vs. SSC in NIRS 2. The hypothesis that rcSO2 values are not different during SSC as compared to incubator care or resting periods only refers to those within person comparison. The comparison between NIRS 1 and NIRS 2 is not relevant for the hypothesis. Also, I could not find the results of the 3 Mann-Whitney U tests in the manuscript, neither for care vs. SSC in NIRS 2.
    3. 4, lines 137-140: As you do not test the difference between both conditions using all the available data (or at least you did not report p-values here), you should refrain from statements such as “lower”, as the slight descriptive differences could just be due to measurement errors.
    4. 4, lines 140-142: “When comparing the periods of each child during SSC and resting time in NIRS 1 the median values for rcSO2 were slightly different…” I did not understand from this phrasing that you are referring to the usage of only one median value for each child here. I only understood this information from comparing the values to Table 3b.
    5. Table 3b: I would rather add the sample size per infant to part a of the table, as it is more relevant there in my opinion. It took me some time to understand Table 3 in its current form. The structure is rather confusing.
    6. Figure 2: All 3 depicted comparisons in the figure should be Wilcoxon signed-rank tests, as they depict comparisons within the same sample. Yet, according to your notes below the figure you conducted Mann-Whiteney U tests, which should only be applied to a comparison of the same period between NIRS 1 and NIRS 2.
    7. Figure 2: In the notes it says “Additionally, the 95% confidence interval of the difference between the periods is shown in square brackets.” I could not find these CI in the figure.
    8. What is your cut-off value for “marginal differences”?
  7. 7, lines 197-198: As the comparison between care and rest was an exploratory analysis, the results should be labelled as such in the discussion.
  8. Some minor grammatical errors:
    1. In the abstract, it should be “were comparable during SSC” instead of “were comparably during SSC”.
    2. First paragraph: “…additional monitoring of the regional…” instead of “…additional monitoring with of the regional…”

Author Response

Reviewer 2:

The study investigates cerebral tissue oxygenation in preterm infants as a biomarker for brain vulnerability. It is used to determine the safety of skin-to-skin contact during the first 120h of life in preterm infants. Therefore, cerebral tissue oxygenation is compared between periods of resting time, skin-to-skin contact, and nursing care. The study has the potential to alleviate concerns in the medical field towards early skin-to-skin contact, as the data suggests no substantial differences in cerebral tissue oxygenation. Given the multitude of beneficial effects of early skin-to-skin contact, the study provides a significant contribution to the field.

Yet, there are some aspects which need to be addressed, particularly concerning the statistical analysis:

  1. I was wondering why you did not choose to combine the data of both studies. Besides the gap in data collection, I could not identify any meaningful differences in the data collection for both studies. The gestational age in NIRS 2 was slightly lower, but as you do not investigate the effect of gestational age, I would not see this as a reason not to combine the data. Should I have missed the reason why the data cannot be combined, please explain it in more detail.

Ad 1) We thank the reviewer for the advice and additionally analysed combined datasets of both studies. We revised the results accordingly: „In a combined analysis of both studies the median values for rcSO2 showed a significant difference between SSC and resting times (median and quartiles in %: 81 [76; 86] vs. 82 [77; 87]; OR with 95% confidence interval from GLMM: 1.178 [1.103; 1.253], p<0.0001; revised Table 3b,c, new Figure 2). We observed no difference between SSC and nursing care (81 [76; 86] vs. 81 [76; 86]; 1.011 [0.930; 1.092], p=0.7869). The predictions based on the GLMM for different gestational ages, gender and time periods are shown in new Figure 2. New Figure 3 demonstrates the intraindividual differences for each infant according to median values during the periods I-III.

  1. You explore the difference in regional cerebral oxygenation saturation between resting time and handling care. While this is an interesting and important question in my opinion, it is not introduced in the introduction. The topic should at least be added as an exploratory hypothesis, stating that you will look at the comparison, even though you have no clear expectations concerning it.

Ad 2) We revised the introduction section accordingly: „In observational studies with infants at a gestational age of 26 0/7 – 31 6/7 and 24 0/7 to 28 6/7 weeks, respectively, we tested our hypotheses that (i) rcSO2 values are not different during SSC as compared to resting periods and (ii) rcSO2 values are lower during handling time as compared to resting periods of infants.

  1. The statement in Heading 3.3 is different from your statement within the text. As you do not test the difference between both conditions, I do not see the statement in Heading 3.3 justified by the analysis. If you did test it, please report the results fully.

Ad 3) Thjank you, revisions were made accordingly.

  1. What is given in parenthesis in Table 1? I’m assuming the numbers in brackets are confidence intervals, but neither is specified.

Ad 4) Revisions were made accordingly. Data are given as numbers (percentage) or median (quartile 1; quartile 3).

  1. In my opinion, the chosen statistical methods are not appropriate for the data structure or the research question. When you’re using all the available data, you do not currently account for dependencies in the data. When you only use the median, you lose the majority of your power. In addition, classical significance testing cannot be used to make statements such as “There is no difference in rcSO2 between SSC and resting”, you can only say “The hypothesis that no difference in rcSO2 between SSC and resting exists cannot be rejected”, which is not the same. I also don’t understand why you use non-parametric tests designed for non-normally distributed data. Theoretical considerations and your descriptive statistics suggest that your dependent variable is normally distributed. Given your hypothesis and your data structure, I would suggest Bayesian multilevel models (also called mixed models). The advantage here is that all your available data could be included, while also controlling for the dependencies in the data. Your data is hierarchical with two levels, assessments at Level 1 and participants at Level 2. The time period (incubator vs. SSC vs. care) is a Level 1 predictor as a characteristic of the assessment. And you could also control for aspects like gestational age, which is a Level 2 predictor as a characteristic of the participant (if you add this predictor to your analysis, z-transform it first as a version centering at the grand mean, see Enders & Tofighi, 2007). Both and more possible confounding variables (e.g., gender) could be included in the analysis at the same time.As you list SPSS as a software you are already using, I would suggest JASP to conduct Bayesian multilevel models. It is as easy to use and openly available without charge. It also offers reasonable priors which you can use, should you not want to specify your own. You can use “Bayesian Linear Mixed Models” if your data is normally distributed as I think and “Bayesian Generalized Linear Mixed Models” if they are not. JASP will automatically dummy code your time period variable for the analysis.
    The advantage with this method over yours is that you’ll receive credibility intervals, which can be interpreted to make statements as “There is no difference in rcSO2 between SSC and resting”. You also do not lose any power and you can address your hypothesis with one test instead of multiple. You also can use your data as it is, without adding an artificial error.

Ad 6) We thank the reviewer for the important statistical advice. In order to address the issues we modified the statistical methods and performed the requested multilevel model as frequentist generalized linear mixed model (not Bayesian) and not in JASP but in R. The data are given in table 3c. The methods section was edited accordingly: „Descriptive statistical analyses were performed for clinical parameters. For testing and visualization of the infants receiving SSC, the two studies were merged due the overlapping gestational ages between the studies and the effect the gestational age has on the rcSO2 values. To test for differences in infants who received SSC, we used a generalized linear mixed model (GLMM) for beta distributed response, with corresponding logit link, and the infant connected with time interval as random effect. Gender, gestational age, and study were used as adjustment. Due to the limited value range of rcSO2 between 15% and 95% the values of rcSO2 were rescaled to the uniform measure. The model cannot handle values of 1 and therefore another transformation ((x * (n – 1) + 0.5)/n) was done where n is the sample size and x the value to transform. For the prediction the transformations were reversed. The type I error level was set to 0.05. To test the differences between pairs of periods with SSC two tests were performed, and the Bonferroni-adjusted type I error level was accordingly set to 0.025 for an overall significance level of 0.05. We used R version 4.1.2 (The R Foundation, Austria, Vienna) together with SPSS 26.0 data analysis package (IBM Copl, New York USA) for all computations and visualizations. Plots were generated using R package ggplot2 (3.3.5) and R-function glmmTMB together with predict.glmTMB from package glmmTMB (1.1.2.3) for the generalized linear mixed model and the corresponding prediction.

Kathrin Hanke and Tanja Rausch (statistician) have contributed equally to the manuscript.

7) What is your cut-off value for “marginal differences”? 7, lines 197-198: As the comparison between care and rest was an exploratory analysis, the results should be labelled as such in the discussion.

Ad 7) We clarified these aspects accordingly under discussion: “We found a median 1% difference of rcSO2 values during SSC episodes and resting time periods, while hypoxic episodes (<55%) as per definition of the Safe-BOOS-C trial were rare events [12]. In an exploratory analysis of resting periods versus nursing care periods no differences in rcSO2 values were demonstrated.”

8.Some minor grammatical errors: In the abstract, it should be “were comparable during SSC” instead of “were comparably during SSC”. First paragraph: “…additional monitoring of the regional…” instead of “…additional monitoring with of the regional…”

Ad 8) Revisions were made accordingly.

With all the revisions made we now hope that our manusctript will be suitable for publication in CHILDREN and look forward to hearing from you.

Thank you very much for your time and efforts.

Sincerely yours,

Kathrin Hanke, Tanja Rausch and Christoph Härtel on behalf of all co-authors

Round 2

Reviewer 1 Report

Thanks for revising the manuscript and making it clearer.

Reviewer 2 Report

The authors comprehensively addressed my previous concerns. I especially appreciate the addition of Figure 2, as it provides the reader with an easy overview of the relevant results. The updated analyses strengthen the results and conclusions drawn from the data.

I have no further suggestions and believe that the manuscript would be a valuable addition to the scientific field.